# Methodology of Specialist Physicians Training: From Traditional to e-Learning

**DOI:** 10.3390/ijerph17207681

**Published:** 2020-10-21

**Authors:** Juan Chaves, Antonio A. Lorca-Marín, Emilio José Delgado-Algarra

**Affiliations:** 1Public Company for Health Emergencies (EPES), 21003 Huelva, Spain; juanmiguel.chaves@juntadeandalucia.es; 2Department of Integrated Didactics, University of Huelva, 21007 Huelva, Spain; antonio.lorca@ddcc.uhu.es

**Keywords:** science education, blended learning, electronic learning, medical education, advanced life support, ALS, scientific research, quasi-experimental research, comparative study

## Abstract

Different studies show that mixed methodology can be effective in medical training. However, there are no conclusive studies in specialist training on advanced life support (ALS). The main objective of this research is to determine if, with mixed didactic methodology, which includes e-learning, similar results are produced to face-to-face training. The method used was quasi-experimental with a focus on efficiency and evaluation at seven months, in which 114 specialist doctors participated and where the analysis of the sociodemographic and pre-test variables points to the homogeneity of the groups. The intervention consisted of e-learning training plus face-to-face workshops versus standard. The results were the performance in knowledge and technical skills in cardiac arrest scenarios, the perceived quality, and the perception of the training. There were no significant differences in immediate or deferred performance. In the degree of satisfaction, a significant difference was obtained in favour of the face-to-face group. The perception in the training itself presented similar results. The main limitations consisted of sample volume, dropping out of the deferred tests, and not evaluating the transfer or the impact. Finally, mixed methodology including e-learning in ALS courses reduced the duration of the face-to-face sessions and allowed a similar performance.

## 1. Introduction

Cardiac arrest continues to be a major health problem, causing a significant number of deaths in Europe [1]. Knowledge of advanced life support techniques (ALS) generates an indisputable benefit by improving survival prognosis. ALS courses are essential in the training of specialist physicians due to aspects such as insufficient training in cardiopulmonary resuscitation in faculties [2], the difficulty of obtaining adequate training with professional exercise [3], the relevance of training periodically that guarantees a correct application of the techniques [4,5], or the need to update protocols and procedures according to international recommendations. Different studies over the years have shown that e-learning can be effective in the field of health science education [6,7,8,9,10]. According to the 2015 Recommendations of the European Resuscitation Council (ERC) [11], there is evidence that supports blended teaching models (independent electronic teaching incorporated into a short, instructor-led course) and promotes the implementation of new teaching technologies based on Information and Communication Technologies (ICT) [4]. In works comparing the traditional face-to-face and mixed methodology, which include ICT, several authors [8,12] found no significant differences in the qualifications between the two pedagogical approaches. However, in ALS courses, Perkins et al. (2012) [12] found a lower performance in simulated scenarios of cardiac arrest in students who use e-learning in relation to entirely face-to-face courses, a fact that is not confirmed by Thorne et al. (2015) [7] with a larger sample. On the other hand, Huynh (2017) [8] reports higher satisfaction in medical students in a mixed learning environment compared to a traditional classroom. In Spain, Castillo’s studies on mixed training (2018 and 2020) [10,13] can be highlighted, although both studies are related to basic life support (BLS) and automatic external defibrillation (AED) training. In the current circumstances, it seems necessary to optimize time in classrooms and invest it in putting into practice what has been learned in simulated cardiac arrest scenarios and, thus, training and executing techniques that can save lives [4,14]. Our study, in the context of the training of specialist doctors, aims to contrast the didactic efficiency of ALS training with mixed methodology, which includes e-learning, versus traditional training, both after finishing the training and in the follow-up evaluation at seven months and, additionally, to evaluate the level of satisfaction of students and estimate the perception of the students about their own training. In each of the objectives set, the scientific hypotheses define similar results with both methodological approaches. As a main conclusion, it is established that the mixed methodology with e-learning can have a didactic efficiency analogous to traditional training in the acquisition of knowledge and technical skills in ALS training.

## 2. Materials and Methods 

### 2.1. Design Overview 

From a positivist research paradigm, the study was framed in quantitative methodology. The method used was quasi-experimental with an efficiency approach, with a before–after design (pre-test–post-test) with a non-equivalent control group, which was complemented with a follow-up evaluation (post–post). This quasi-experimental study compared the results of ALS training between a control group (T) with traditional training (self-study period supported by a paper-based manual and face-to-face classes, both theoretical and practical) and an experimental cohort (EL) with mixed methodology, which included e-learning and face-to-face workshops (self-study period tutored through an e-training platform, supported by interactive and multimedia didactic material, and face-to-face practical workshops). Each group (control and experimental) included students from three training actions: the T cohort courses were developed for 22 contact hours (two and a half days) and the EL cohort courses in 12 contact hours (one day and a half). The students enrolled in the training actions were resident specialist doctors who were in their first year of specialty in various Andalusian hospitals; as part of their study plan, they are required to take an ALS course. There was no random assignment of participants to groups; rather, groups of already established students were used without altering them, groups whose individual assignment corresponded to the teaching units of the hospitals. The control group was formed with 54 subjects and the experimental group with 60. Finally, 52 and 58 students completed the courses, respectively. All of the face-to-face phases of the training actions were carried out in the same institution and facilities, using the same didactic and simulation resources.

### 2.2. Definition of Variables

The learning method was established as an independent variable, considering the application of mixed didactic methodology as an intervention that included online learning. For this reason, the independent variable had two values: the mixed teaching methodology and the traditional one, considering it as an independent task situational variable. The study’s dependent variables were established as academic performance (acquired knowledge and practical skills), retention after seven months, student satisfaction, and how able the student felt to perform cardiopulmonary resuscitation (perception of the training itself). Table 1 specifies the variables.

Ten confounding sociodemographic variables were identified that could have some influence on the dependent variables. These variables are: age, sex, the type of hospital in which the residency (specialization) was carried out, the faculty of where the bachelor’s degree was studied, the number of years to study the degree, time elapsed since the completion of the bachelor’s degree, completion in advance of an ALS course, professional practice of medicine prior to residency, basic computer knowledge at the user level, and interest in taking this type of course. These confounding variables from the subjects were taken into consideration to analyse whether they were distributed homogeneously between the groups and, therefore, to know if their differential effects could be equalized in order to determine the equivalence between the control and experimental groups with respect to at least these characteristics.

Another confounding variable was related to the teachers who gave the training. For this reason, as a partial control of this variable outside of the objective of the research and to know if there were significant differences between the groups, the evaluation values of the teachers by the students were taken into account.

### 2.3. Intervention

In both groups, the ALS of ERC recommendations were followed. In the control group, the training was imparted in a conventional way. In these groups and four weeks in advance, the students received the ALS book, the manual published by the National Plan of the Spanish Society of Intensive Care, Critical Care and Coronary Units (SEMYCIUC). After this self-study phase, the face-to-face phase was carried out. In this phase, the instructors taught ten hours of theoretical classes and twelve hours of practical workshops and cardiac arrest simulations. In the experimental group, the students had access to the tele-training platform for four weeks, allowing them to exercise control over the sequence and pace of learning, content, and time of dedication. Additionally, they were provided with a compact disc recognized by the Spanish Council for Cardiopulmonary Resuscitation (CERCP) and published by the Iavante Foundation (Foundation for Technological Advancement and Professional Training) from the Health Council of the Andalusian Regional Government. This included ERC recommendations and contained schematics, images, algorithms, and videos in which key advanced resuscitation techniques and procedures were explained step-by-step. Table 2 specifies the teaching materials included in the e-learning phase.

After the e-learning phase, the face-to-face phase was carried out. In this phase, instructors taught twelve hours of practical workshops and cardiac arrest simulations. Both the facilities and the material for the practical workshops and simulations were the same for both groups. The workshops were developed in both cohorts in small groups, with a maximum teacher–student ratio of 1:8. 

### 2.4. Data Collection Instruments: Validity and Reliability

Data collection was carried out using the following instruments:-Survey on sociodemographic data (age, sex, interest in taking the training action, academic data, work experience, and computer skills). This survey was completed anonymously when the groups were trained and before starting the training actions.-Knowledge test with 25 questions and five answers for each one. This test was carried out by the students in three successive moments: it was identical in the tests before (pre), after (post), and during follow-up evaluation at seven months (post–post). This consisted of the same test for the control (T) and experimental (EL) group.-Checklist to evaluate technical skills in simulation situations, both in the post-test and in the follow-up evaluation. The rubric included nine domains using specific indicators that included complex techniques and application of algorithms, in addition to assessing security and solvency in the application of the protocols. Table 3 shows the evaluated technical skills.

According to the number of inadequately executed techniques, four levels of achievement were subsequently established (Table 4). 

In case of discrepancy between the observers, the student’s achievement level was determined assuming the most unfavourable.
-Teacher evaluation survey by students: it was completed after the post-test. It included the following items: mastery of knowledge, clarity of transmission, adaptation to group needs, mastery of technical and didactic resources, and compliance with the established schedule.-Surveys on satisfaction and perception of their training: these surveys were completed by the students after the tests and follow-up evaluation. The satisfaction survey was carried out using a four-item Likert scale (from 1 (not adequate) to 4 (very adequate)). Although controversial [15], a four-item Likert scale was chosen to avoid the effects of data collected with a bias towards central values [16]. The item “General assessment of the course” was only included in the post survey. For the items “Do you feel capable of performing cardiopulmonary resuscitation (CPR)?” and “I would recommend the course”, the assessment proposed to the students was from 0 to 10.

Due to the relevance of the validity and reliability of the measurement instruments, the following data collection instruments were selected:-Test on knowledge in ALS: questions from examinations of ALS courses recognized by CERCP were used.-Practical skills checklist: this checklist was reviewed by three ALS instructors from the SEMICYUC National Cardiopulmonary Resuscitation Plan, with recognition from the ERC. Two of these instructors were the observers who completed the checklist in the post-test and follow-up tests on all subjects simultaneously and independently. The instructor-observers did not participate as course teachers. The instructor-observers were selected among the members of the teaching team for their professional and teaching experience, but they did not participate as teachers in the courses and did not interfere in the development of the simulation situations. To verify the concordance and consensus of the observations of both instructors, statistical measures of agreement between reviewers were used.

### 2.5. Methodological Limitations

The study design was quasi-experimental, and subjects were not randomly assigned to the experimental and control groups. It had a control group, which gave it greater robustness and reasonable control over most sources of disability. Additionally, the identification and analysis of confounding variables allowed us to know if they were homogeneously distributed among the cohorts, checking the equivalence of the groups with respect to these possibly related characteristics. Another limitation was the sample volume and the loss of subjects in the follow-up evaluation, since the dropouts weaken the statistical quality and, therefore, the retention findings have to be taken with greater caution. The loss of subjects could be conditioned by the non-obligatory nature of deferred participation. In any case, it was found in each group that there were no significant differences in the results of the posttest tests (knowledge and technical skills) between the subsets of subjects who did carry out the follow-up evaluation and those who did not.

### 2.6. Statistical Tests for Data Analysis

Qualitative variables are presented as absolute frequency and percentage (%). The quantitative variables are exposed with the three indices that determine the data distribution: size (n), mean (m), and standard deviation (SD). The statistical comparison between the control group T and the experimental group EL was made for the quantitative variables with the Student’s t-test (independent and related samples), taking into consideration the assumption of equality of variances (Levene’s test) and the normality of the distribution (reaffirmed with the non-parametric Kolmogorov–Smirnov test). For qualitative variables, the comparison between the groups was made with Pearson’s Chi-square, and the Mann–Whitney U test was used to compare two independent samples with ordinal variables. On the other hand, to know the correlation and the level of agreement between observers, the Spearman correlation coefficients, Gamma coefficient, and Cohen’s kappa index were obtained. In this study, *p* values less than 0.05 were considered statistically significant differences. SPSS version 19 software was used for statistical analysis.

## 3. Results

The training actions were initiated by 114 subjects, but were completed by 110 students: 52 in the control group (T) and 58 in the experimental group (EL). Twenty students of the 17.54% of those assigned to the courses did not have sociodemographic data or qualifications from the previous test (pre-test). Regarding sociodemographic information, we had data from 51 subjects of the T group and 43 of the EL group.

### 3.1. Analysis of the Homogeneity of Groups

The objective of the initial analysis was to know if the study cohorts had similar characteristics. Taking into consideration the sociodemographic variables studied—including the one referring to the interest of the students in carrying out the training action—and the results of the diagnostic evaluation tests (pre-test), it was possible to consider, with sufficient confidence, the homogeneity of both groups for these variables analysed. Additionally, similar values were found in the evaluation of the students to the teachers, which could have partially influenced the control of this other strange variable identified. No statistically significant differences were obtained in the strange variables: age, sex, type of hospital in which the residency was carried out (specialization), faculty where the bachelor’s degree was studied, number of years in studying the degree, time elapsed since completion of undergraduate degree, completion of an advanced life support course, medical practice prior to residency, basic computer knowledge at the user level, and interest in taking this type of course. 

The percentage of resident physicians in the age range between 24–29 years was 82.35% in the control group and 86.05% in the experimental group, and 60.78% and 65.12% were women in the T and EL groups, respectively. In Figure 1, the age groups and the percentages of each group are detailed.

Regarding origin, 50.98% of the students in group T carried out residency in a regional hospital, 33.33% in specialty hospitals, and 15.69% in regional hospitals. In the EL group the percentages were 53.49% in regional, 25.58% in specialty hospitals, and 20.93% in regional. The independence hypothesis could not be ruled out, because the probability associated with Pearson’s Chi-square statistic was 0.653 (*p* > 0.05). For the variable regarding the university in which the students studied medicine, 75.55% of the control group reported that it was the Malaga faculty, with the same faculty of medicine being 55.81% of the experimental group. In this case, the observed significance level associated with the Chi-square statistic was 0.254. Additionally, neither the Contingency Coefficient nor Cramer’s V confirmed a significant association. In Figure 2, the universities where the students studied are grouped together.

Likewise, no statistically significant difference was obtained regarding the previous completion of an ALS course: 76% of the control group students had undergone training in advanced resuscitation techniques prior to the research study compared to 64% of the group experimental. In the evaluation of teachers by students, the items analysed were: knowledge mastery, clarity of transmission, adaptation of group needs, mastery of technical and didactic resources, and compliance with the established schedule. Table 5 shows the descriptive statistics of the results obtained. There were no statistically significant differences between the groups.

### 3.2. Results of Knowledge Tests

In relation to the knowledge pre-test, in group T the mean was 12.76 correct questions with a standard deviation (SD) of 2.654, compared to 12.82 correct questions (SD = 2.975) in EL. No significant differences were found between both groups (*p* = 0.920). After training, group T obtained an average of 21.94 correct questions in the knowledge post-test (SD = 1.720) compared to 21.84 (SD = 2.016) (*p* = 0.787). Accepting an α error of 0.05 in the bilateral contrast with 52 students in the control group and 58 in the experimental group, the power of the hypothesis contrast was 80% (β error = 0.20) to detect the difference between means of a “Unit” in the knowledge post-test (a difference of one question out of the 25 that made up the test) as statistically significant. It was considered as the minimum difference with practical significance.

The results of the pre- and post-tests in the study groups were compared to determine the intrinsic gain by applying the Hake Index (g) (fraction of maximum gain possible per instruction). For each group, the percentage of correct answers for the pre-test and post-test was analysed, and the total learning gain was found. The normalized mean gain (g) is presented, applying: g = G/(100–I), being the average percentage of gain G = O–I, where I is the average percentage of correct questions at the beginning of the course (pre-test) and O the average percentage at the end of the course (post-test). The denominator (100–I) represents the percentage of the maximum possible profit that students can make. The results are shown in Table 6.

The gain in total learning using the Hake Index was similar in both groups (T: 0.75 and EL: 0.74); these data revealed a high gain in learning (*g* > 0.7).

In the deferred phase at seven months, the number of participants in the tests decreased significantly: the control group (T) was made up of 37 students (28.85% lost) and the experimental group (EL) was made up of 29 (50% lost). In group T, an average of 20.14 correct questions (SD = 2.323) was obtained compared to 20.72 in group EL (SD = 2.313) (*p* = 0.310). No significant difference was found between the groups, although, due to the abandonment produced, the result must be considered with prudence. Table 7 shows the comparison of the knowledge tests according to the teaching methodology.

Knowledge declined statistically significantly in both groups, but the decrease in retention was significantly more marked in the control group: the mean of the related differences (t with paired data, post follow-up test) was 1.595 (95% CI: 0.864–2.325; *p* = 0.000) vs. 1.034 (95% CI: 0.250–1.819; *p* = 0.012). This decrease in retention and the relationship that the test results had with each other are made visible in Figure 3, which shows the position of the test means of each group, together with the limits of the confidence interval (95%) that corresponded to those averages. 

### 3.3. Results of Technical Skills Test 

In the global assessment of skills in advanced resuscitation techniques, no significant differences were found between both groups at the end of the training actions (post-tests); the mean of techniques performed improperly was 3.19 (SD = 1.645) in the T group compared to a mean of 3.03 (SD = 1.706) in the EL group (significance level = 0.623). After seven months (post post-tests), although with no statistically significant differences between the study cohorts, a lower number of errors or techniques were observed in the experimental group, as can be seen in Table 8. In this case, it seemed necessary to take into account that the control group was made up of 37 students and the experimental group of 29 and, Therefore, accept the result with due caution.

When comparing the post follow-up technical skills tests, a non-significant increase in the mean number of errors or techniques performed improperly was observed in both groups (mean of related differences: t with paired data): in the traditional group = 0.270 (95% CI: −0.409 to 0.949; *p* = 0.425) vs. 0.172 in the EL group (95% CI: −0.635 to 0.980; *p* = 0.425). In the mixed methodology group, Figure 4 shows in error bars the position of the means of the inappropriate techniques of each group, together with the limits of the confidence interval (95%) that corresponded to these means (individual intervals for individually considered means).

If the performance data in technical skills competencies is presented according to the established achievement levels, in the post tests, 53% of the students in the experimental group were framed in the satisfactory or excellent levels (achievement levels 3 and 4) vs. 44% in the control group, and in the follow-up tests (post–post), 55% in the EL compared to 38% in the T group. Table 7 shows the percentages of the two groups according to the levels of achievement achieved both in the test later and in the follow-up evaluation.

The findings of both groups presented in Table 9 are graphically represented in Figure 5, post-test and Figure 6, follow-up evaluation, in which the relationship between them can be observed for each of the achievement levels achieved.

Because the evaluations of these practical skills were carried out by two independent observers, and assuming a similar level of experience, it was pertinent to measure the level of agreement that existed between both judges. With this objective, the Spearman correlation coefficient and the Gamma coefficient—measures of linear association for ordinal variables—and the Cohen’s kappa index were obtained in terms of the agreement on performance according to the unacceptable and limit levels of achievement (1 and 2) vs. satisfactory and excellent levels (3 and 4). The measures of agreement in the post and post–post tests confirmed significant linear relationships. This happened with the Spearman and Gamma rho coefficients (Table 10), useful tests in this case because they collected measures of association, taking advantage of ordinal information (achievement levels), which measured designed for nominal data overlook [17].

On the other hand, apart from the statistical significance of the kappa index (0.000) that allowed us to conclude that there was an agreement greater than that expected by chance, the value reached in the evaluations of the post skills tests (*k* = 0.561) reflected an agreement among judges that was moderately good (moderate), although in the post post-tests (*k* = 0.612) a substantial agreement (substantial) could be considered [18].

### 3.4. Results on Satisfaction Level

Regarding the quality perceived by the students in the training actions of the two groups, it was possible to affirm that there was a different level of satisfaction (general assessment of the course) and that this difference was statistically significant using an ordinal scale. Thus, in the traditional group (T) a mean of 3.58 (SD = 0.572) was obtained, compared to a mean of 3.30 (SD = 0.566) in the mixed methodology group (EL), with a *p* = 0.012. When using the association measures for ordinal data, we obtained a significance of 0.008 with the Mann–Whitney U test, which corroborated the statistically significant difference found. Figure 7 shows the count of students in each group, ordered according to their degree of satisfaction with the training action.

In addition to the difference found in general satisfaction, a significant difference was found in the item “The contents are adapted to the objectives of the course”: group T: 3.55 (SD = 0.566) and group EL: 3.30 (SD = 0.572); *p* = 0.021. There were no significant differences between the two groups regarding issues related to the development of the face-to-face phase: facilities, assessment of practical workshops, or facilitation of learning by using the didactic methodology of robotic simulation. Likewise, no statistically significant difference was found in the degree of satisfaction relative to the overall duration of the course: group T: 2.88 (SD = 0.807) and group EL: 2.82 (SD = 0.636); *p* = 0.691. However, as shown in Figure 8, significant differences were obtained regarding the duration of the face-to-face phase (significance level = 0.001 with Chi-square): 43.6% of the students in the group with mixed methodology with e-learning would have preferred a face-to-face phase with more hours. On the contrary, 30.8% of the students in the group with traditional methodology would have preferred a face-to-face phase with fewer hours.

Regarding the assessment of the usefulness of the material provided (traditional: manual; E-learning: CD) and usefulness of the theoretical sessions (control group) and tele-training platform (experimental group), both items asked in the questionnaire of the deferred phase, no statistically significant differences were found. Regarding the item “I would recommend the course”, no significant differences were observed between the group means: 8.48 in the T group vs. 8.42 in the EL group (score from 0 to 10 points).

### 3.5. Results of Perception about Training

Both in the post survey and after seven months (post–post), the students were questioned about whether they “feel capable of performing cardiopulmonary resuscitation (CPR)”. The results obtained did not reflect statistically significant differences; although, the means presented higher values for the experimental cohort, as shown in Table 11.

## 4. Discussion

In this quasi-experimental study with an efficiency approach, traditional training in ALS was compared to a blended training that included e-learning. From the results obtained, it was not apparent that, in the training actions with mixed didactic methodology, a lower performance was achieved in terms of the acquisition of knowledge and technical skills. This confirmed the proposed scientific hypothesis. The intrinsic gain was determined by applying the Hake Index (g), analysing the percentage of correct answers for the pre-test and post-test, and a high learning gain was obtained in both groups (group T = 0.75 and group EL = 0.74). In the deferred phase of the study, in which attendance was not mandatory, the control group (T) was made up of 37 students (28.85% lost) and the experimental group (EL) 29 (50% lost). Due to dropouts at the follow-up assessment, the similar retention finding at seven months was viewed with caution. In this follow-up evaluation, knowledge declined significantly in both groups; however, the decrease occurred more markedly in the traditional group: 1.595 (95% CI: 0.864 to 2.325; *p* = 0.000) vs. 1.034 (95% CI: 0.250 to 1.819; *p* = 0.012). Regarding the mean of inadequate techniques, it was higher in the traditional group (3.65 vs. 2.90; significance level = 0.081). Additionally, with the t-test with paired data (mean of the related differences), a non-significant increase in the mean number of errors or techniques performed improperly was detected with respect to post-tests: in the control group = 0.270 (95% CI: −0.409 to 0.949; *p* = 0.425) vs. 0.172 (95% CI: −0.635 to 0.980; *p* = 0.425) in the experimental group. Regarding the level of satisfaction of students questioned by the general assessment of the course using a Likert scale of four items (from 1 (not at all adequate) to 4 (very adequate)), in the traditional group there was an average of 3.58 (SD = 0.572) vs. 3.30 (SD = 0.566), with a significance level (sig.) of 0.012 (sig. = 0.008 with the Mann–Whitney U test). This statistically significant difference refuted the proposed scientific hypothesis. However, when it was asked (score from 0 to 10) whether “I would recommend the course”, no significant differences were found in this item: group T = 8.48 (SD = 0.183) vs. group EL = 8.42 (SD = 0.164) (*p* = 0.808). In both the post–post survey and the deferred survey (post–post), when asked whether “you feel capable of performing cardiopulmonary resuscitation (CPR)”, the results obtained did not reflect significant differences; in any case, the means presented higher values for the mixed methodology group: 7.49 (SD = 0.133) vs. 7.33 (SD = 0.199) (*p* = 0.495; without assuming equal variances) in the after test, and 6.34 (SD = 0.167) vs. 6.03 (SD = 0.228) (*p* = 0.288) in the follow-up evaluation. The design of our study was quasi-experimental, to which the control group conferred robustness, although this was not equivalent (without individual randomization), which fundamentally differentiates it from controlled randomized experimental studies. In order to determine whether in the control and experimental groups there were possible variables other than the intervention that could interfere with the results, the sociodemographic characteristics identified as potentially confounding variables were analysed and, thus, allowed us to be able to know if they appeared equally in both groups, which would allow its effect to be considered controlled. Additionally, the subjects of both groups performed a pre-test, which provided information on the comparability of the groups. Each group (control and experimental) was made up of students from three training actions: resident medical specialists who were in their first year of specialty and who had established in their study plan the obligation to take an ALS course for the research. The groups were not altered, but the established ones were used whose enrolment is usually carried out by the teaching units of the hospitals, being considered in the design phase groups very similar to each other, not showing signs of systematic errors or biases in the analysis. No statistically significant differences were obtained in the sociodemographic variables (possible confounding variables). Likewise, in the evaluation of the teachers by the students, there were no statistically significant differences in the items analysed, which could partially influence the control of this other confounding variable identified. Regarding the results of the knowledge pre-test, there were no significant differences; data that reaffirm with due caution the homogeneity of the groups.

The instruments for data collection received special attention. Using the questionnaires to assess the knowledge of a member entity of the Spanish Council for Cardiopulmonary Resuscitation, the rubric for the evaluation of technical skills was reviewed by three ALS instructors from the SEMYCIUC National Plan and its verification was carried out—for all the subjects that formed the study groups—by the same two medical instructors that were independent observers who simultaneously evaluated the students of the training actions and did not participate in the teaching.

The literature review confirmed the introduction of e-learning, increasingly broad, in multiple areas of education; also, in the context of medicine [19,20], in the field of life support training and virtual tools, studies were found regarding BLS and AED. However, those related to ALS were very few, highlighting those published by Perkins et al. [12] and Thorne et al. [7], and no study was found in Spain. It is possible that the complexity of the domains of knowledge and skills to be developed in advanced training could have made it difficult to approach research studies. Certainly, the most ambitious work published was that of Perkins et al. (2012) [12], a multicentre randomized experimental study with a control group in which 3732 healthcare professionals and medical students from the United Kingdom and Australia participated, which suffered from a high percentage of dropouts (24%), a fact that could have influenced the results, as the author himself mentioned. In this study, no significant differences were found in the students’ grades, although there was a lower performance in scenarios of cardiac arrest in the experimental group. The work of Thorne et al. (2015) [7] included students from 1350 courses delivered by the Resuscitation Council for eighteen months across the UK. The students voluntarily enrolled in the different training activities, resulting in 18,952 taking the face-to-face course and 8218 students taking the course with mixed methodology with e-learning. This study had a pre-test–post-test design for the knowledge tests. The students who enrolled in the e-learning group obtained higher marks both in the pre-test and in the post-test, the differences being significant in both cases. Likewise, the students of the e-learning courses achieved slightly higher scores in the practical simulations, the difference being statistically significant as well. The study by Thorne et al. concluded by affirming the equivalence in the learning of ALS with both didactic methodologies and that courses with mixed methodology with e-learning should be promoted. As a continuation of this work, Thorne et al. [21] found that the time spent accessing e-learning materials did not affect the results of the course. In our setting, Castillo’s studies (2018 and 2020) [10,13] can be highlighted, although both are studies related to basic life support training. The work of Castillo et al. (2018) [13] had the objective of comparing the immediate efficacy and at six months regarding BLS and AED, using standard and mixed methods (self-study video). The participating subjects were 129 first-year medical and nursing students who were randomly assigned to the control and experimental groups. In this study, it was concluded that the mixed methodology provides the same or higher levels of knowledge and skills than the traditional one, both after completing the course and at six months. The study by Castillo et al. published in 2020 [10] is a randomized experimental study, which compares the results of training in BLS and AED between a control group with face-to-face training and an experimental group with mixed methodology and performs a follow-up evaluation at nine months; 89 second-year dental degree students participated, obtaining similar results in both cohorts. As far as we know, our work is the first study with an efficiency approach carried out in Spain where didactic methodologies in ALS training are compared and a follow-up evaluation is carried out. As in outstanding works by other authors, the results of this study suggest that with a mixed methodology that includes e-learning, results similar to those obtained with a traditional training entirely face-to-face can be achieved. Additionally, the satisfaction of the students and the perception of the students about their own training were analysed. Regarding the perception of their competences, Carrero et al. (2010) [22] did analyse this aspect, but related to the BLS. Consistent with the findings reported by other authors [10,22,23], the level of satisfaction of the participants was high in both cohorts. Despite this, our hypothesis of finding differences due solely to chance was refuted and, therefore, our consideration that multimedia material and online content would be highly attractive to students and an advantage to reduce face-to-face activity was not fully backed. In a study by Lockey et al. (2015) [24], ALS courses with mixed methodology were well received by most but not all participants. This found that the individual learning styles of students influenced their reaction to course materials. In the work by Abdullah et al. (2019) [25], 58% of students considered that e-learning could replace face-to-face teaching, while 85% reported that e-learning should be used as a complement to conventional education.

In our study on training in ALS, a follow-up evaluation (post–post) was included at seven months. However, there was, as in most studies [10], a loss of subjects that weakens the statistical evidence. Other authors [10,26,27] have carried out the delayed evaluation in the interval of 6 to 12 months, although in the BLS domain, an element that could be of interest and that we have not found explicit in related studies is related to the consistency of the intraobserver or interobserver measurement in the evaluation of technical skills competencies in complex simulation scenarios. In the multicentre randomized study by Perkins et al. (2012) [12], two instructors completed a rating scale; however, the degree of agreement between them is not analysed. In the design phase of our study, the importance of evaluating the agreement between observers was considered, which to a certain extent was indirectly ratified by the findings obtained. Thus, two observers, very experienced instructors, had to evaluate extremely standardized techniques with well-established international quality criteria. Following the classical criteria of Landis and Koch (1977) [18], a moderately good agreement (*k* = 0.561) in post skills tests and substantial agreement (k = 0.612) in follow-up evaluations were obtained, without prejudice to the fact that the cited criteria may have fallen into disuse due to their empirical support [28]. Undoubtedly, higher values of the kappa index were expected in our study. Regarding the measures of linear association for ordinal variables (Spearman and Gamma coefficients), a strong correlation and association was obtained between judges. In any case, we consider that in future studies, special care must be taken to manage disagreement before starting the study, given the importance of analysing the consistency of the measurements. 

Blended learning, defined as the combination of traditional face-to-face learning and asynchronous or synchronous e-learning, has grown rapidly and is being used more and more widely in education. In a systematic review and meta-analysis that included 56 articles [20]—as Cook et al. [19] previously found that, compared to no intervention, blended learning appears to have a positive effect constant, and is more effective or at least as effective as non-blended learning for the acquisition of knowledge in the health professions. She also referred to the great heterogeneity of approaches in the courses with mixed methodology that address the BLS. The electronic interventions—which do not exceed two hours—are more limited, and there is very frequent use of videos or interactive electronic computer programs for the self-instruction phase [10,29], although studies have been published in this regard [30,31,32,33], in more complex training, such as training in the ALS. Surely, we must continue to advance in the knowledge about which tools for e-learning are the most effective for the acquisition of different competences. The SARS-CoV-2 pandemic that originated in the spring of 2020, the massive confinement, and the closure of educational institutions for face-to-face training should urge the educational community as a whole to a calm but broad reflection of online training at all educational levels. This will also be a fundamental field to be addressed by educational research professionals. A key element will consist of the teachers’ own training in e-learning teaching techniques, since they have to know the tools in depth, act as learning facilitators, and have the skills to motivate students in their self-training, which will result in the effectiveness of this methodology.

In this study, data has been collected, essentially, on performance and retention related to the acquisition of knowledge and technical skills, but no data have been obtained regarding the acquisition of non-technical skills (soft skills). The transfer of learning to the clinic, impact on mortality, or costs have been analyzed. There is evidence that the mixed methodology provides significant savings for the provider [34,35]. In other research studies, it could be feasible to approach the comparison of didactic methodologies in ALS training contemplating the acquisition of non-technical skills. In this context, our experience has focused on the development of training actions with advanced simulation, in which we incorporate the training in realistic environments of relationship skills, such as effective communication, leadership, decision-making, and work capacity as a team in stressful situations. Thus, using innovative didactic methodologies of clinical simulation with the use of advanced robots and didactic dramatization tools, and with a scenery as complete and truthful as possible, plus debriefing sessions supported by video recordings, should form the basis of the advanced training of health professionals [36,37,38].

The findings of this work suggest that ALS training through mixed didactic methodology can have a didactic efficiency analogous to the traditional one in the accessible population, first-year resident interns who specialize in hospitals in the province of Malaga. Although it is not conclusive, and with due caution, it is possible that the mixed methodology could be an alternative to the traditional courses in the white population, a population made up of first-year specialty residents of hospitals in the autonomous community of Andalusia.

## 5. Conclusions

In this comparative study on the training methodology for specialist physicians, the results obtained through mixed teaching methodology (which included online learning) were similar to those obtained with traditional classroom methodology, both in terms of immediate performance after completing the course and at the seven-month follow-up evaluation. Consequently, these findings confirm the proposed scientific hypotheses and suggest that, in the acquisition of knowledge and technical skills in ALS training, the mixed methodology with e-learning could have a didactic efficiency analogous to the traditional one, at least in the accessible population studied. Regarding the level of student satisfaction, the results showed a slightly higher score in the traditional group (3.58 compared to 3.30), although they refuted the research hypothesis by rejecting the statistical hypothesis, since the difference found was statistically significant. Despite this, no significant difference was found when asked whether they would recommend the course. In this work, after completing the training actions and seven months after the intervention, the perception that physicians expressed about their own training in cardiopulmonary resuscitation was similar in both groups, with no statistically significant differences; however, higher scores were achieved in the experimental group. The mixed methodology approach, which included e-learning, reduced the duration of the face-to-face sessions, and, from the study findings, it was not apparent that a lower performance was achieved in terms of the acquisition of knowledge and technical skills, both immediately after the course and in the deferred assessment. Therefore, although not conclusively, the mixed methodology could be a valid didactic alternative in the ALS training of internal physicians residing in Andalusia.

## Figures and Tables

**Figure 1 ijerph-17-07681-f001:**
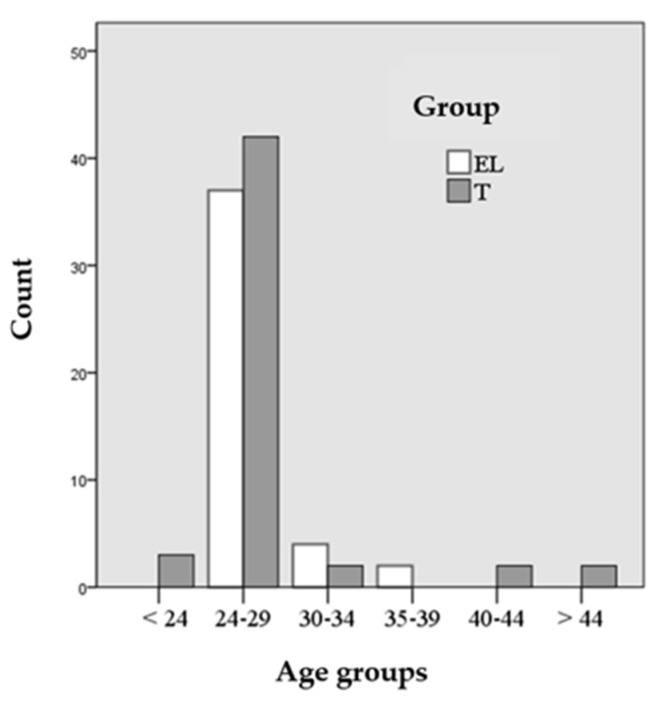
Age groups of students.

**Figure 2 ijerph-17-07681-f002:**
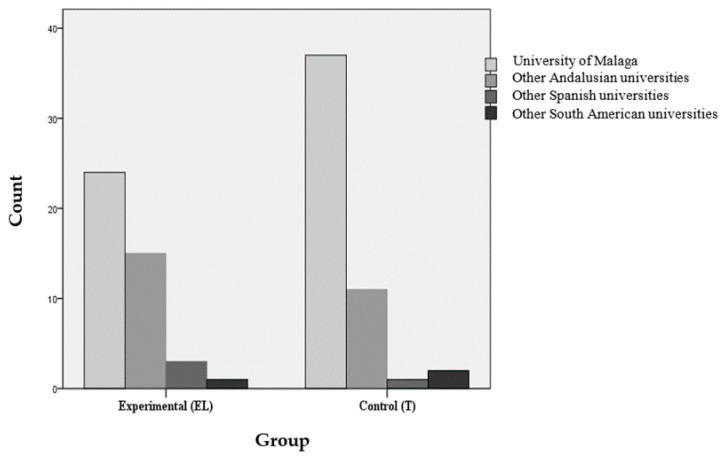
Alumni alma mater.

**Figure 3 ijerph-17-07681-f003:**
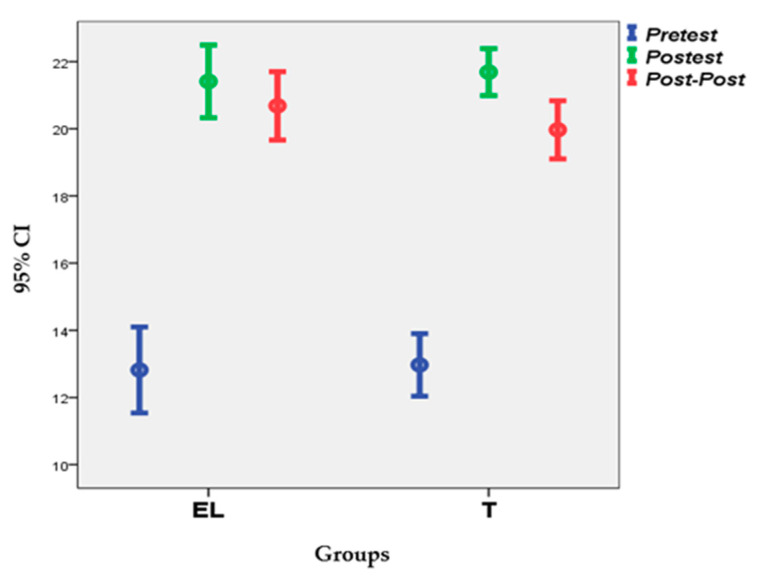
Knowledge test means and confidence intervals (95%).

**Figure 4 ijerph-17-07681-f004:**
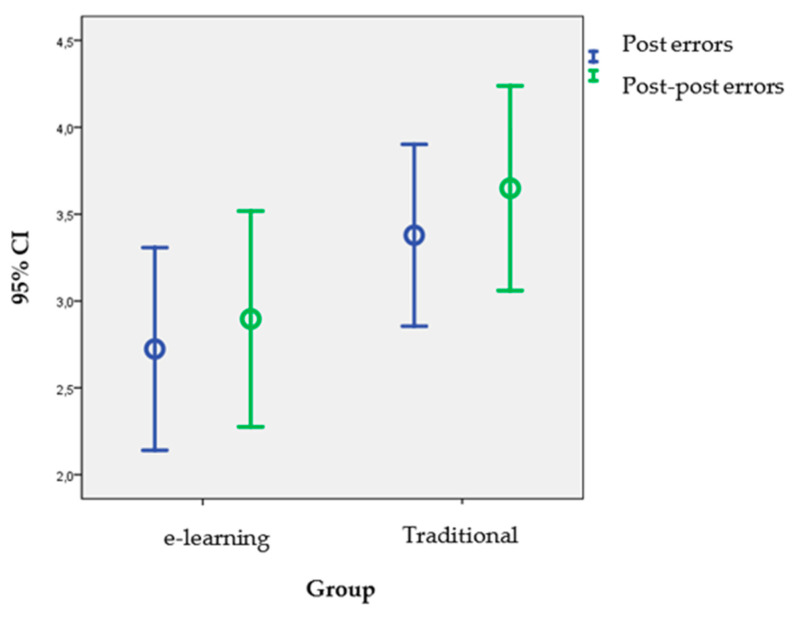
Inadequate technical means and confidence intervals (95%).

**Figure 5 ijerph-17-07681-f005:**
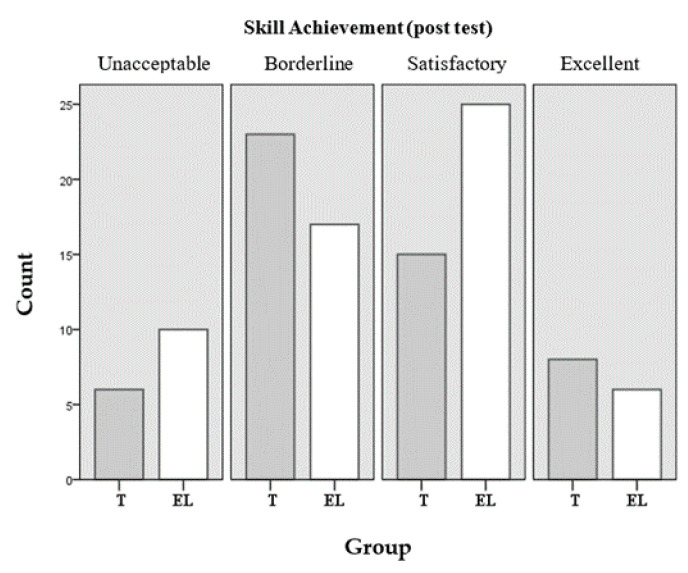
Achievement levels in technical skills (post-test).

**Figure 6 ijerph-17-07681-f006:**
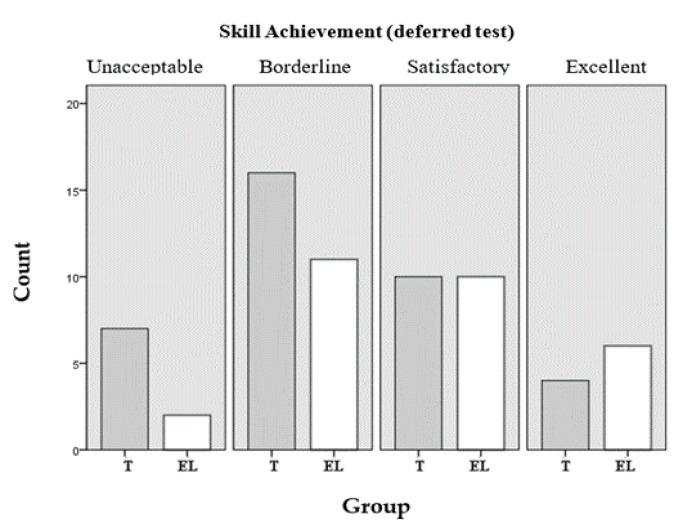
Achievement levels in technical skills (deferred test).

**Figure 7 ijerph-17-07681-f007:**
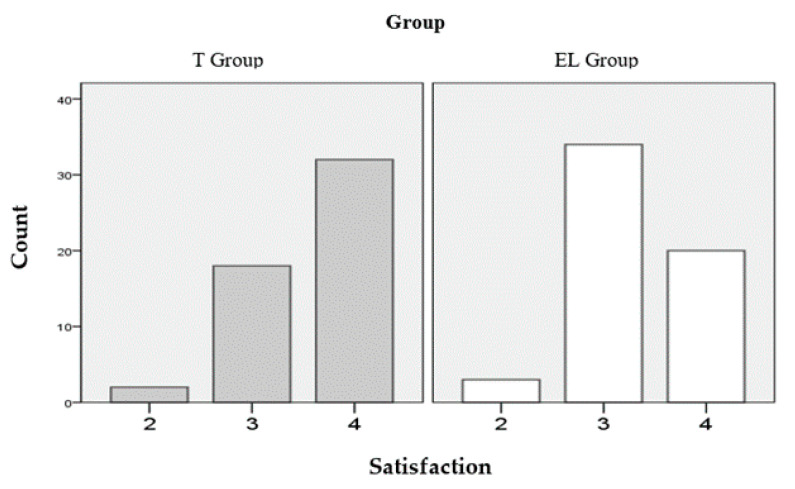
Level of satisfaction (general assessment) after taking the course (Likert scale of four items from 1 = not at all adequate to 4 = very adequate).

**Figure 8 ijerph-17-07681-f008:**
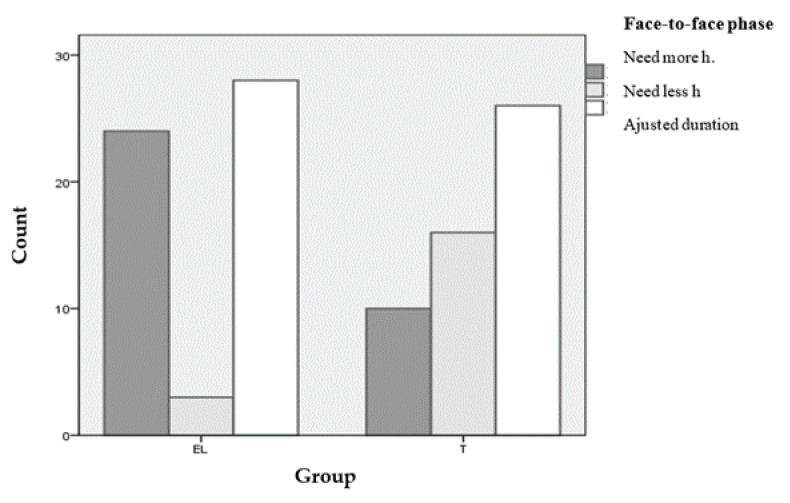
Duration of the face-to-face phase: student preferences.

**Table 1 ijerph-17-07681-t001:** Operational definition of the dependent variables.

Variables	Operational Definition
Knowledge	Tests score
	(post and post–post)
Technical skills	Simulation score
	(post and post–post)
Satisfaction	Surveys score
	(post)
Perception about their training	Surveys score
	(post and post–post)

**Table 2 ijerph-17-07681-t002:** Teaching materials included in ALS e-learning.

Theoretical Models	Six Interactive Models
Self-test	
Clinical scenarios of cardiac arrest	
Simulations of peri-arrest arrhythmias	Tachyarrhythmias
	Bradyarrhythmias
Videos of twelve techniques related to:	Defibrillation
	Cardioversion
	Upper airway adjuncts and suction
	Supraglottic airway
	Intubation
	Ventilation
	Cricothyrotomy
	Central venous access catheter

**Table 3 ijerph-17-07681-t003:** Technical skills assessed in post and post–post-tests.

Domains	Indicators
High-quality CPR (manual chest compressions)	Hand position
	Compressions depth
	Compressions rate
	CPR sequence
Defibrillation (manual defibrillator)	Interruptions to chest compressions
	One shock and immediate chest resume compressions
	Adequate shock energy levels
Ventilation (bag-valve-mask)	Airway adjuncts
	Normal chest rise
	Give oxygen
Advanced airway management: Supraglottic airway	Interruptions to chest compressions
	Provides effective ventilation
Advanced airway management: Endotracheal intubation	Interruptions to chest compressions
	Confirm correct tube position
	Provides effective ventilation
Intravenous access and drug therapy	Give IV adrenaline 1 mg every 3–5 min
ALS treatment algorithm: Shockable rhythms (VF/pVT)	Check patient/call for help/CPR 30:2
	Attach defibrillator/monitor. Minimise interruptions
	Assess rhythm/deliver shock
	Continuous compressions when advanced airway in place
	Amiodarone 300 mg after 3 shock
ALS treatment algorithm: Non-Shockable rhythms (PEA/Asystole)	Check patient/call for help/CPR 30:2
	Attach defibrillator/monitor. Minimise interruptions
	Assess rhythm/NO shock
	Continuous compressions when advanced airway in place
	Recognize and treat reversible causes
Confidence and self-control	Apply the algorithms showing confidence and self-control

Note: CPR = cardiopulmonary resuscitation; IV = intravenous; VF = ventricular fibrillation; pVT = pulseless ventricular tachycardia; PEA = pulseless electrical activity.

**Table 4 ijerph-17-07681-t004:** Achievement levels according to the number of improper techniques or procedures.

Achievement Level	Number of Improper Techniques
Nivel 1. Unacceptable	6–9
Nivel 2. Borderline	4–5
Nivel 3. Satisfactory	3–2
Nivel 4. Excellent	0–1

**Table 5 ijerph-17-07681-t005:** Evaluation of teachers by students in experimental groups (EL) and control groups (T).

	Group	*n*	Mean (SD)	*p*
Knowledge	T	52	9.31 (0.78)	0.36
	EL	39	9.15 (0.81)	
Clarity	T	52	8.94 (0.92)	0.61
	EL	39	8.85 (0.84)	
Adaptation	T	52	8.87 (0.99)	0.68
	EL	39	8.95 (0.86)	
Technical and didactic resources	T	52	8.96 (0.97)	0.95
	EL	39	8.95 (0.89)	
Compliance with schedule	T	52	9.13 (0.99)	0.73
	EL	39	9.21 (0.89)	

Note: SD = Standard deviation; *p*-values < 0.05 are considered statistically significant.

**Table 6 ijerph-17-07681-t006:** Hake Index values.

Group		Pre-Test	Post-Test	Hake Index (g)
Control	Subjects (n)	50	52	0.75
(Traditional)	Mean of correct questions	12.76	21.94	
	% Correct questions	51.04%	87.76%	
Experimental	Subjects (n)	44	58	0.74
(e-Learning)	Mean of correct questions	12.82	21.84	
	% Correct questions	51.28%	87.36%	

**Table 7 ijerph-17-07681-t007:** Knowledge test results: pre, post, and post–post.

		Test of Knowledge		
		Pre	Post	Post–Post
Control (T)	*n*	50	52	37
	mean	12.76 (2.65)	21.94 (1.72)	20.14 (2.32)
Experimental (EL)	*n*	44	58	29
	mean	12. 82 (2.98)	21.84 (2.02)	20.72 (2.31)

Note: Knowledge is expressed in the number of correct questions on the 25 question multiple choice tests. The values expressed in parentheses correspond to the standard deviation of the mean.

**Table 8 ijerph-17-07681-t008:** Descriptive statistics of the post–post technical skills test improperly executed.

	Group	*n*	Mean	Standard Deviation	Standard Error of Mean
Improperly technics	T	37	3.65	1.767	0.291
post–post	EL	29	2.90	1.633	0.303

Note: Significance level = 0.081 for equality of means of T and EL groups.

**Table 9 ijerph-17-07681-t009:** Achievement level reached in technical skills.

Achievement Level	Number of Inappropriate Techniques	Skills Tests “After” (Post)		Skills Tests “Deferred” (Post–Post)	
		Control (T)	Exper. (EL)	Control (T)	Exper. (EL)
		*n* = 52	*n* = 58	*n* = 37	*n* = 29
**Nivel 1** Inacceptable	6–9	11.54%	17.24%	18.92%	6.90%
**Nivel 2** Borderline	4–5	44.23%	29.31%	43.24%	37.93%
**Nivel 3** Satisfactory	2–3	28.85%	43.10%	27.03%	34.48%
**Nivel 4** Excellent	0–1	15.38%	10.35%	10.81%	20.69%
		100.00%	100.00%	100.00%	100.00%

**Table 10 ijerph-17-07681-t010:** Linear association measures: agreement between reviewers and technical skills.

		Spearman Rho	Gamma
	Value	0.621	0.74
Post agreement	Approx. sig.	0.000 *	0
	*n* (valid cases)	110	110
	Value	0.823	0.976
Post–Post agreement	Approx. sig.	0.000 *	0
	*n* (valid cases)	66	66

Note: Sig. = significance level. (*) Based on the normal approximation.

**Table 11 ijerph-17-07681-t011:** Students’ perception of their own training.

	Group	*n*	Mean	Standard Deviation	Standard Error of Mean	*p*
**Post survey**	Control (T)	52	7.33	1.438	0.199	0.495
	Experimental (EL)	57	7.49	1.002	0.133	
**Post** **–** **Post survey**	Control (T)	37	6.03	1.384	0.228	0.288
	Experimental (EL)	29	6.34	0.897	0.167	

Note: Assessment proposed to students from 0 to 10 points.

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
