# Peer review of "Methodology of Specialist Physicians Training: From Traditional to e-Learning"

_ijerph, 2020, doi:10.3390/ijerph17207681_

Round 1

Reviewer 1 Report

Thank you for the opportunity to review the article "Methodology of Specialist Physicians Training: From 2 Traditional to e-Learning"

1) It is worth adding that the effectiveness of e-learning trainings depends largely on the form of communication. This results in discrepancies among researchers of BLS, ALS, etc. courses. There is evidence that FILMS is an unfavorable form of knowledge transfer. I ask the authors to write more details on the teaching materials included in ALS e-learning (were there self-tests, interactive materials, educational games, etc.).

2) I propose a linguistic correction of the text. Phrases such as "Another strange variable (...)" are unusual.

3) Please describe in more detail the selection of experts and their verification during the evaluation of practical tasks. The description of the authors indicates the use of the Delphi method. It belongs to the group of heuristic methods which use the knowledge, experience and opinions of experts in a given field to make decisions.

4) Please specify exactly what activities were assessed in the practical tasks.

5) Please specify the reasons for such a significant difference in the number of people taking part in the test after 7 months.

6) The authors used a rather unusual Likert scale (4 points) in the satisfaction assessment. Could the authors indicate why the 5-point scale was not chosen?

7) References contain 25 items, but only 5 of them concern contemporary research (from the last 3 years). I believe that when describing modern e-learning solutions, one should indicate the current literature. I am asking the authors to include the following articles in the manuscript:

1) Boczkowska K, Bakalarski P, Sviatoslav M, et al .. The importance of e-learning in professional improvement of emergency nurses. Crit. Care Innov. 2018; 1 (1): 16-24.
DOI: 10.32114 / CCI.2018.1.1.16.24

2) Leszczyński P, Gotlib J, Kopański Z, et al .. Analysis of Web-based learning methods in emergency medicine: randomized controlled trial. Archives of medical science: AMS. 2018; 14 (3): 687.
DOI: 10.5114 / aoms.2015.56422

3) Adhesive T, Jayamaha AR. Knowledge of the in-hospital resuscitation algorithm among medical staff of selected hospital departments. Crit. Care Innov. 2019; 2 (2): 9-16.
DOI: 10.32114 / CCI.2019.2.2.9.16

4) Sobolewska P, Pinet-Peralta LM. Use of the educational mobile applications by emergency medical services personnel. Crit. Care Innov. 2019; 2 (2): 25-31.
DOI: 10.32114 / CCI.2019.2.2.25.31

5) Leszczyński P, Charuta A, Łaziuk B, et al .. Multimedia and interactivity in distance learning of resuscitation guidelines: a randomized controlled trial. Interactive Learning Environments, 2018; 26 (2): 151-162.
DOI: 10.1080 / 10494820.2017.1337035

6) Reid D, Sim M, Beatty S, et al .. Pre-hospital advanced life support resuscitation – a curriculum for pre-hospital education. Australasian Journal of Paramedicine, 2020; 17 (1).
DOI: 10.33151 / ajp.17.757

Author Response

Dear reviewer, thank you very much for the recommendations that will improve the final article. We send the report of changes included.

Thank you for the opportunity to review the article "Methodology of Specialist Physicians Training: From 2 Traditional to e-Learning"

  • It is worth adding that the effectiveness of e-learning trainings depends largely on the form of communication. This results in discrepancies among researchers of BLS, ALS, etc. courses. There is evidence that FILMS is an unfavorable form of knowledge transfer. I ask the authors to write more details on the teaching materials included in ALS e-learning (were there self-tests, interactive materials, educational games, etc.).

Table 2, line 117-121

  • I propose a linguistic correction of the text. Phrases such as "Another strange variable (...)" are unusual.

Line 100 “Other confounding variable…”

  • Please describe in more detail the selection of experts and their verification during the evaluation of practical tasks. The description of the authors indicates the use of the Delphi method. It belongs to the group of heuristic methods which use the knowledge, experience and opinions of experts in a given field to make decisions.

Line 168-171.

  • Please specify exactly what activities were assessed in the practical tasks.

Table 3, line 140-142

Lines 181-184, Line 445-446

  • The authors used a rather unusual Likert scale (4 points) in the satisfaction assessment. Could the authors indicate why the 5-point scale was not chosen?

Lines 154-156

  • References contain 25 items, but only 5 of them concern contemporary research (from the last 3 years). I believe that when describing modern e-learning solutions, one should indicate the current literature. I am asking the authors to include the following articles in the manuscript:

Boczkowska K, Bakalarski P, Sviatoslav M, et al .. The importance of e-learning in professional improvement of emergency nurses. Crit. Care Innov. 2018; 1 (1): 16-24.
DOI: 10.32114 / CCI.2018.1.1.16.24

Reference 30, line 698-699. Cited in text, line 569

Leszczyński P, Gotlib J, Kopański Z, et al .. Analysis of Web-based learning methods in emergency medicine: randomized controlled trial. Archives of medical science: AMS. 2018; 14 (3): 687. DOI: 10.5114 / aoms.2015.56422

Reference 31, line 700-701. Cited in text, line 569

3Adhesive T, Jayamaha AR. Knowledge of the in-hospital resuscitation algorithm among medical staff of selected hospital departments. Crit. Care Innov. 2019; 2 (2): 9-16. DOI: 10.32114 / CCI.2019.2.2.9.16

Reference 5, line 641-642. Cited in text, line 35

Sobolewska P, Pinet-Peralta LM. Use of the educational mobile applications by emergency medical services personnel. Crit. Care Innov. 2019; 2 (2): 25-31.
DOI: 10.32114 / CCI.2019.2.2.25.31

Reference 33, line 705-706. Cited in text, line 569

Leszczyński P, Charuta A, Łaziuk B, et al .. Multimedia and interactivity in distance learning of resuscitation guidelines: a randomized controlled trial. Interactive Learning Environments, 2018; 26 (2): 151-162. DOI: 10.1080 / 10494820.2017.1337035

Reference 32, line 702-704. Cited in text, line 569

Reid D, Sim M, Beatty S, et al .. Pre-hospital advanced life support resuscitation – a curriculum for pre-hospital education. Australasian Journal of Paramedicine, 2020; 17 (1). DOI: 10.33151 / ajp.17.757

Reference 9, line 648-649. Cited in text, line 37

Other references were included:

Abal, F. J. P.; Auné, S. E.; Lozzia, G. S.; Attorresi, H. F. Funcionamiento de la categoría central en ítems de Confianza para la Matemática. Revista Evaluar 2017, 17(2), 18-31. Available online:  https://revistas.unc.edu.ar/index.php/revaluar (accessed on 24.09.2020).

Reference 15, line 664-666. Cited in text, line 155

Bisquerra, R.; Pérez-Escoda, N. ¿Pueden las escalas Likert aumentar en sensibilidad? Revista d’Innovació i Recerca en Educació 2015, 8(2), 129-147.

Reference 16, line 667-668. Cited in text, line 156

Thorne, C.J.; Lockey, A.S.; Kimani, P.K.; Bullock, I.; Hampshire, S.; Begum-Ali, S.; Perkins, G.D. Advanced Life Support Subcommittee of the Resuscitation Council (UK). e-Learning in Advanced Life Support-What factors influence assessment outcome? Resuscitation 2017, 114, 83-91.

Reference 21, line 676-677. Cited in text, line 513

Lockey, A.S.; Dyal, L.; Kimani, P.K.; Lam, J.; Bullock, I.; Buck, D.; Davies, R.P.; Perkins, G.D. Electronic learning in advanced resuscitation training: The perspective of the candidate. Resuscitation 2015, 97, 48-54.

Reference 24, line 684-685. Cited in text, line 536

Abdullah, A.A.;Nor, J.; Baladas, J.; Hamzah, T.M.; Kamauzaman, T.H.; Noh, A.Y.; Rahman, A. E-learning in advanced cardiac life support: outcome and attitude among healthcare professionals. Hong           Kong journal of emergency medicine 2019, 1-6.

Reference 25, line 686-688. Cited in text, line 539

O’Doherty, D.; Dromey, M.; Lougheed, J.; Hannigan, A.; Last, J.; McGrath, D. Barriers and solutions to online learning in medical education – an integrative review. BMC Medical Education 2018, (18), 130.

Reference 34, line 707-708. Cited in text, line 582

George, P.P.; Ooi, C.K.; Leong, E.; Jarbrink, K.; Car, J.; Lockwood, C. Return on investment in blended advanced cardiac life support training compared to face-to-face training in Singapore. Proceedings of Singapore healthcare 2018, 27(4), 234-42

Reference 35, line 709-711. Cited in text, line 582

Reviewer 2 Report

This is a quasi-experimental study looking at the impact of a blended learning approach to ALS training as opposed to a traditional face-to-face versions. The outcomes addresses include knowledge acquisition, skills acquisition and participant satisfaction. The authors conclude that the blended learning approach is no worse than the traditional approach for knowledge and skills, although there was a preference for face-to-face with the traditional course. 

The methodology is clear and the authors go into great detail to justify the non-matched control group approach. This is perfectly valid. Having said that, the paper would have been supported by a single traditional Table that clearly documented the demographics in the two groups. The detail about "strange factors" felt as if it could be more briefly described in tabular format as well. 

I was unclear if this study was performed in one teaching centre only, and how many courses were run in total. were the same teaching faculty used for all courses? The numbers in both arms of the study were relatively small and this should be declared as a limitation.

In the results section, it is clearly stated in the Skills section that dropout rates in the post post-test group meant that the results should be viewed with caution. this same statement needs to be included for the knowledge results section, as there is no indication if a 50% dropout included those with predominantly more or less knowledge. 

The first section of the discussion could be substantially condensed. Lines 410-468 contain considerable repetition of the results section, as opposed to true discussion.

The literature review contains papers relating to BLS training, which is not directly as relevant as those for ALS training. There are several papers surrounding blended learning for ALS training that are not referenced that could maybe enrich the discussion section specifically relating to ALS courses instead:

  • Lockey et al.electronic learning in advanced resuscitation training: the perception of the candidate. Resuscitation 2015;97:48-54
  • Thorne et al. e-learning in advanced life support - what factors influence assessment outcome? resuscitation 2017;114:83-91
  • Arithra Abdullah et al.e-learning in advanced cardiac life support: outcome and attitude among healthcare professionals. Hong Kong journal of emergency medicine 2019
  • George et al. return on investment in blended advanced cardiac life support training compared to face-to-face training in Singapore. proceedings of Singapore healthcare 2018;27(4):234-42

Ultimately, this adds to the paucity of published work supporting the concept of a blended learning approach to ALS education. Overall, I felt that the manuscript could be a bit more concisely worded but that the messaging is important.

Minor Points

First two lines of the abstract are inaccurate - why are you referring to e-learning alone as an opening statement when the study relates to Blended Learning? also, I would contend that the Thorne paper and also the Arithra Abdullah paper are conclusive that this approach can be used.

line 173 - what is meant by "20 students of the 17.54% of those 173 assigned to the courses do not have sociodemographic data or qualifications from the previous test"

line 128 - should be table 2, not 3

table 3 - would benefit from p values

line 517 - typo. some additional words that appear out of context

line 537 - odd abbreviation - suggest use full term blended learning 

Author Response

Dear reviewer, thank you very much for the recommendations that will improve the final article. We send the report of changes included.

This is a quasi-experimental study looking at the impact of a blended learning approach to ALS training as opposed to a traditional face-to-face versions. The outcomes addresses include knowledge acquisition, skills acquisition and participant satisfaction. The authors conclude that the blended learning approach is no worse than the traditional approach for knowledge and skills, although there was a preference for face-to-face with the traditional course. 

The methodology is clear and the authors go into great detail to justify the non-matched control group approach. This is perfectly valid. Having said that, the paper would have been supported by a single traditional Table that clearly documented the demographics in the two groups. The detail about "strange factors" felt as if it could be more briefly described in tabular format as well. 

Thank you for your recommendation; However, we didn´t include a table because in the new version we have 11 of them.

I was unclear if this study was performed in one teaching centre only, and how many courses were run in total. were the same teaching faculty used for all courses? The numbers in both arms of the study were relatively small and this should be declared as a limitation.

Lines 72-80, line 179

In the results section, it is clearly stated in the Skills section that dropout rates in the post post-test group meant that the results should be viewed with caution. this same statement needs to be included for the knowledge results section, as there is no indication if a 50% dropout included those with predominantly more or less knowledge. 

Lines 181-184, line 282

The first section of the discussion could be substantially condensed. Lines 410-468 contain considerable repetition of the results section, as opposed to true discussion.

According to this recommendation, non-essential elements were deleted and modified.

The literature review contains papers relating to BLS training, which is not directly as relevant as those for ALS training. There are several papers surrounding blended learning for ALS training that are not referenced that could maybe enrich the discussion section specifically relating to ALS courses instead:

  • Lockey et al.electronic learning in advanced resuscitation training: the perception of the candidate. Resuscitation 2015;97:48-54

Reference 24, line 684-685. Cited in text, line 536

  • Thorne et al. e-learning in advanced life support - what factors influence assessment outcome? resuscitation 2017;114:83-91

Reference 21, line 676-677. Cited in text, line 513

  • Arithra Abdullah et al.e-learning in advanced cardiac life support: outcome and attitude among healthcare professionals. Hong Kong journal of emergency medicine 2019

Reference 25, line 686-688. Cited in text, line 539

  • George et al. return on investment in blended advanced cardiac life support training compared to face-to-face training in Singapore. proceedings of Singapore healthcare 2018;27(4):234-42

Reference 35, line 709-711. Cited in text, line 582

Ultimately, this adds to the paucity of published work supporting the concept of a blended learning approach to ALS education. Overall, I felt that the manuscript could be a bit more concisely worded but that the messaging is important.

Minor Points

First two lines of the abstract are inaccurate - why are you referring to e-learning alone as an opening statement when the study relates to Blended Learning? also, I would contend that the Thorne paper and also the Arithra Abdullah paper are conclusive that this approach can be used.

Line 11

line 173 - what is meant by "20 students of the 17.54% of those 173 assigned to the courses do not have sociodemographic data or qualifications from the previous test"

It will be delete in the last version.

line 128 - should be table 2, not 3

New tables were included and that error was reviewed.

table 3 - would benefit from p values

They number of tables changed. Considering this, we included p values in table 5, lines 254-255 and table 11, lines 436-437.

line 517 - typo. some additional words that appear out of context

line 537 - odd abbreviation - suggest use full term blended learning 

lines 560, 563, 564.

Round 2

Reviewer 2 Report

I am happy that the issues raised in my review have been addressed

Author Response

I upload the latest version with some specific adjustment in writing. Thanks for the review.
